# The *Brucella* Effector Protein BspF Regulates Apoptosis through the Crotonylation of p53

**DOI:** 10.3390/microorganisms11092322

**Published:** 2023-09-15

**Authors:** Ruiqi Lin, Ang Li, Yuzhuo Li, Ruitong Shen, Fangyuan Du, Min Zheng, Jinying Zhu, Jingjing Chen, Pengfei Jiang, Huan Zhang, Jinling Liu, Xiaoyue Chen, Zeliang Chen

**Affiliations:** 1Key Laboratory of Livestock Infectious Disease, Ministry of Education, Shenyang Agricultural University, Shenyang 110866, China; ruiqilinland@163.com (R.L.); liang000624@outlook.com (A.L.); s1105370014@163.com (R.S.); fangyuandu@126.com (F.D.); 15585020531@163.com (M.Z.); jean_zz512@163.com (J.Z.); 15776544198@163.com (J.C.); pengfeijiang1998@163.com (P.J.); ljlfreeone@163.com (J.L.); xiaoyuechen@syau.edu.cn (X.C.); 2Key Laboratory of Ruminant Infectious Disease Prevention and Control (East), Ministry of Agriculture and Rural Affairs, Shenyang 110866, China; 3Department of Epidemiology, School of Public Health, Sun Yat-sen University, Guangzhou 510275, China

**Keywords:** *Brucella*, BspF, GNAT, decrotonylation, intracellular survival

## Abstract

The *Brucella* type IV secretion system (T4SS) can promote the intracellular survival and reproduction of *Brucella*. T4SS secretes effector proteins to act on cellular signaling pathways to inhibit the host’s innate immune response and cause a chronic, persistent *Brucella* infection. *Brucella* can survive in host cells for a long time by inhibiting macrophage apoptosis and avoiding immune recognition. The effector protein, BspF, secreted by T4SS, can regulate host secretory transport and accelerate the intracellular replication of *Brucella*. BspF has an acetyltransferase domain of the GNAT (GCN5-related N-acetyltransferases) family, and in our previous crotonylation proteomics data, we have found that BspF has crotonyl transferase activity and crotonylation regulation of host cell protein in the proteomics data. Here, we found that BspF attenuates the crotonylation modification of the interacting protein p53, which reduces the p53 expression through the GNAT domain. BspF can inhibit the transcription and protein expression of downstream apoptotic genes, thereby inhibiting host cell apoptosis. Additionally, the *Brucella* Δ*bspF* mutant stain promotes apoptosis and reduces the survival rate of *Brucella* in the cells. In conclusion, we identified that the T4SS effector protein BspF can regulate host cell apoptosis to assist *Brucella* in its long-term survival by attenuating crotonylation modification of p53 and decreasing p53 expression. Our findings reveal a unique mechanism of elucidating how *Brucella* regulates host cell apoptosis and promotes its proliferation through the secretion of effector proteins.

## 1. Introduction

*Brucellosis*, caused by the bacterium *Brucella*, is a naturally occurring epidemic disease that affects both humans and animals [1]. This Gram-negative, parthenogenetic intracellular bacterium lacks both pods and spores and is non-motile but contains all the genes necessary for flagellar synthesis [2]. The genus *Brucella* is classified into six species and twenty biotypes based on their antigenicity and biological characteristics. These include the *Brucella melitensis* found in goats, *B. abortus* found in cattle [3], *B. suis* in swine [3], *B. canis* in dogs [4], *B. ovis* in sheep [5] and *B. neotomae* in the desert wood rat [6]. Among these species, *Brucella melitensis* is the most invasive and pathogenic and is most likely to cause outbreaks and epidemics of *Brucellosis* in humans [7]. *Brucellosis* can occur year-round and is present worldwide, with a particularly severe impact on health and the economy. Human infections with *Brucella* are mainly contracted through the consumption of dairy products and undercooked animal meat products that are not properly sterilized, as well as through direct exposure to diseased animals [1,8].

After invasion by *Brucella* into the host organism, the bacteria mainly parasitize macrophages and trophoblast cells [9,10]. *Brucella* produces idiosyncratic niches for replication and persistence through special pathways in host cells [11]. *Brucella* interacts with the cell membrane of macrophages through the lipid rafts and enters the host cells by endocytosis to form *Brucella*-containing vacuoles (BCVs) surrounded by phagocytic vesicles [12]. In an acidic environment, BCVs further increase the expression of the *Brucella* type IV secretion system (T4SS) and can affect virB transcription, while T4SS is closely linked to both BCVs’ development and transport [13,14]. BCVs can take transient fusion with the endosomal and lysosomal to evade the host immune response, which enables the bacteria to survive and replicate [15]. When *Brucella* is transported to the endoplasmic reticulum (ER) of the host cells, the BCVs, at this point, fuse with the ER in a Sar1- and Rab2-dependent manner to obtain sufficient nutrients to replicate intracellularly, which are called replicative BCVs (rBCVs) [16]. After the formation of rBCVs, VirB manipulators interact with the endoplasmic reticulum to attenuate the acidic environment within the vesicles [17]. *Brucella* forms autophagic BCVs (aBCVs) at the end of the endoplasmic reticulum replication, which brings the *Brucella* intracellular cycle to an end, releasing itself from the infected cells and starting to infest new batches of cells [18].

T4SS is one of the most important virulence factors of *Brucella* [19,20]. T4SS can secrete effector proteins that interfere with normal intracellular signaling and contribute to BCVs escaping lysosomal degradation, facilitating *Brucella* to establish replicative niches in the ER [21]. In the current study, at least 15 T4SS effector proteins have been identified, whose functions and actions can influence the mechanisms of *Brucella* proliferation and intracellular survival [22]. The T4SS effector proteins BspA, BspB, and BspF inhibit the secretion of host proteins and promote intracellular replication in *Brucella abortus* [11]. BspB interacts with Conserved oligomeric Golgi (COG) to promote *Brucella* replication, thereby compensating for the impairment of Rab2a’s RicA regulation of *Brucella* replication [23]. The effector protein BspF (BAB1_1948) contains a Gcn5-related N-acetyltransferase (GNAT) family acetyltransferase domain [11,22]. The GNAT protein superfamily is widely present in prokaryotes and is involved in the regulation of stress response, transcription, and metabolism [24]. BspF assists *Brucella* replication in rBCVs by inhibiting vesicle transport in the trans-Golgi network (TGN). The interaction of BspF with the Arf6 GTPase-activating protein (GAP) ACAP1 leads to an imbalance of Arf6/Rab8a-dependent transport in circulating endosomes, resulting in an increase in TGN-associated vesicles in rBCVs and promoting the intracellular replication of *Brucella abortus* [25].

*Brucella* affects cell apoptosis through its virulence factors. Studies have shown that *Brucella* invades the host’s cells and secretes effector proteins that inhibit macrophage apoptosis to evade immune recognition, helping *Brucella* persistently survive in host cells [16]. The effector protein VceC, the first effector of *Brucella* [21], promotes persistent *Brucella* intracellular infection by interacting with Grp78, reducing Grp78 expression, and inhibiting CHOP-induced apoptosis. BspJ is a nuclear regulatory protein secreted by *Brucella,* and it was reported that the BspJ deletion strain promotes apoptosis after cell infestation and reduces the survival of *Brucella* within macrophages [22]. Therefore, the *Brucella* BspJ protein can also inhibit cell apoptosis. These studies suggest that *Brucella* effector proteins play a vital role in regulating cell apoptosis.

Our previous study has reported that the *Brucella* effector protein BspF has de-crotonyltransferase activity and modifies the overall level of crotonylation modifications in host cells. We speculated that BspF may affect protein functions by affecting the level of crotonylation modification in host cells, thereby enhancing *Brucella* replication [26]. Moreover, through the use of mass spectrometry (MS)-based crotonylation modification proteomics data, we have shown that BspF regulates crotonylation at the K351 site of p53, an important protein in the mitochondrial apoptosis pathway. Therefore, we hypothesized that BspF may affect apoptosis by regulating the crotonylation modification of p53 and play an important role in *Brucella* immune evasion and intracellular survival. In this study, we discovered that BspF attenuates the crotonylation modification of p53, leading to a reduced expression in the cells. This regulation of the host cell apoptosis pathway promotes persistent intracellular infection by *Brucella*. Our findings contributed to the exploration of how *Brucella* affects host cell apoptosis and evades the host immune response. They also provide a reference for investigating the host immune response mechanism against intracellular bacterial infections and serve as an important reference for studying non-histone crotonylation modification.

## 2. Materials and Methods

### 2.1. Strains, Cells, and Reagents

In this study, the *Brucella abortus* 2308 (*B. abortus*) wild strain (WT) and 2308 Δ*bspF* mutant strain were obtained from the Institute of Military Sciences (Beijing, China) and were grown in tryptic soy agar (TSA; Takara, Kusatsu, Japan) and tryptic soy broth (TSB; Takara, Kusatsu, Japan).

The *Escherichia coli* strain DH5α (TransGen Biotech) was cultured in Luria–Bertani (LB) medium for gene cloning. HEK-293T cells, derived from human embryonic kidney cells, and HeLa cells, derived from cervical cancer cells, both from our laboratory, were cultured in Dulbecco’s minimal essential medium (DMEM) containing 10% fetal bovine serum (FBS) (Gemin, Woodland, CA, USA) at 37 °C and 5% CO_2_.

### 2.2. Plasmids and Antibodies

The full-length p53 gene was obtained from the PCR reactions (Table 1) using p53-F and p53-R primers (Table 2), with the total cDNA extracted from the cells serving as the template. The PCR instrument was set with the following parameters: predenaturation at 94 °C for 5 min; denatured at 94 °C for 30 s, annealed at 56 °C for 30 s, extended at 72 °C for 30 s, extended at 72 °C for 10 min for 30 cycles, and preserved at 4 °C after the reaction. The PCR-amplified gene was recovered and purified by gel extraction. The p53 gene was cloned into the Flag-tag2B vector (Flag-tag2B-p53). Using the Flag-tag2B-p53 plasmid as a template, the p53 lysine (K)-351 was mutated to alanine (A)-351, and the mutant Flag-tag2B-p53K351A was constructed. The plasmids, pCMV-HA-BspF and pCMV-HA-BspFΔGNAT, have been constructed by our research group. The following antibodies were purchased commercially: anti-Flag-tag mAb (MBL), anti-HA-tag mAb (MBL), anti-p53 mAb (SC-126), anti-Kcr mouse mAb (PTM-502), anti-β-actin mouse mAb (AF0003), anti-Cleaved-caspase-3 rabbit mAb (PTM-7246), anti-AIF rabbit mAb (AF1273), and anti-Bax rabbit mAb (AF1270).

### 2.3. Western Blotting Analysis and Co-Immunoprecipitation Experiments

Using the Endo-Free Plasmid Nidi Kit (Omega Bio-Tek, Guangzhou, China), the Flag-p53 and Flag-p53K351A plasmids were extracted from the bacterial solution. The HEK-293T cells were cultured using 5 × 10^7^ cells/well in 10 cm cell culture dishes in a cell culture incubator at 37 °C with 5% CO_2_. When the cell density reached 80∼90%, Lipofectamine 2000 (VigoFect, Beijing, China) was used to transfect the plasmids (pCMV-HA-BspF, Flag-tag2B-p53) into the HEK-293T cells. After being cultured in the cell incubator for 30 h, the cell culture medium was discarded, and the cells were washed with PBS and lysed with 1 mL RIPA buffer (150 mM NaCl, 50 mM Tris-HCl (pH 7.4), 2 mM Na_2_EDTA, 10% glycerol, 1% NP-40, and 0.1% SDS) supplemented with a protease inhibitor cocktail. After centrifugation for 15 min at 14,000 rpm at 4 °C, 80 µL of the supernatant protein solution was mixed with 5 µL of the 5 × SDS-PAGE protein loading buffer (Beyotime, Shanghai, China) and boiled at 100 °C for 10 min for Western blotting analysis. The 1000 µL remaining supernatant protein solution was supplemented with mouse anti-HA-tag monoclonal antibody (mAb) and mouse anti-Flag-tag mAb (MBL, Kusatsu, Japan) (1:300). The mixture was incubated for 4 hours at 4 °C with rotation. Following this, 40 µL of Protein A + G agarose beads (Beyotime, Shanghai, China) were added and the mixture was incubated overnight at 4 °C with rotation. The beads were washed three times with ice-cold lysis buffer and boiled at 100 °C for 10 min. Finally, the samples were subjected to immunoblotting.

The samples were subsequently separated by SDS-PAGE and transferred onto polyvinylidene difluoride membranes (PVDF) (Millipore, Burlington, MA, USA). After blocking with 5% skimmed milk in TBST, the membrane was incubated with primary antibodies (Mouse anti-HA-tag mAb and Anti-Flag-tag mAb) overnight at 4 °C, followed by incubation with goat anti-mouse IgG (H + L) (Beyotime, Shanghai, China) (1:3000) secondary antibody for 2 h. Finally, the membranes were washed five times in TBST for 5 min. The intensity of the Western blotting band signals was detected with the ultrasensitive ECL chemiluminescence solution (EpiZyme, Shanghai, China) using the Gel Imaging Instrument (Aplegen, Pleasanton, CA, USA) according to the manufacturer’s instructions.

### 2.4. Quantitative Real-Time PCR (qRT-PCR) Assay

We sought to understand the effect of BspF on the transcription of apoptotic genes. The pCMV-HA-BspF was transfected into HeLa cells containing Lipofectamine 2000 (Vazyme, Nanjing, China). After 24 h, RNAiso Plus (Takara, Kusatsu, China) was added to lyse the cells to extract the total RNA, and then we reverse-transcribed it into cDNA using the PrimeScript TM RT reagent Kit with gDNA Eraser (Takara, Kusatsu, China). Using the cDNA as a template, a qRT-PCR was performed using a fluorescence real-time quantitative PCR instrument (Thermo Fisher, Waltham, MA, USA) and ChamQ Universal SYBR qPCR Master Mix (Novozymes, Beijiing, China). The qRT-PCR reaction system is shown in Table 3. The gene transcription levels of Caspase-3, AIF, Bax, Bad, Bcl-2, and p53 were detected by qRT-PCR. The relative expression levels were normalized to β-actin, and the primer sequences are shown in Table 4. Each experiment was performed three times. The fluorescence PCR instrument interface was set as follows: Hold Stage 1: 95 °C for 30 s; PCR Stage 2: 95 °C for 10 s, 60 °C for 30 s, 40 cycles; Melt Curve Stage 3: 95 °C for 15 s, 60 °C for 60 s, 95 °C for 15 s. The relative expression levels of p53, Caspase-3, AIF, Bcl-2 and Bax were calculated by 2^−ΔΔCT^ with β-actin as the reference gene.

### 2.5. Brucella Culture and Cell Infection Assay

*Brucella abortus* S2308 was stored at −80 °C, inoculated into the TSA solid medium, and cultured upside-down in a constant temperature incubator at 37 °C for 2–3 days. A single colony was selected on a TSA plate and placed in a TSB liquid medium at 37 °C with shaking at 180 r/min for 24 h. Then, 1 mL of the bacterial solution was taken and centrifuged at 4500 r/min for 5 min. The TSB was discarded, and the bacterial solution was washed twice with PBS. Finally, the bacteria were re-suspended with PBS and transferred into a transparent glass tube. The concentration of *Brucella abortus* S2308 was determined by McFarland turbidimetry. 

Briefly, HeLa cells were cultured in 6-well plates for 24 h until the cell count reached 1.0 × 10^6^ cells per well. They were then infected with *Brucella abortus* 2308 and *Brucella abortus* Δ*bspF* at a multiplicity of infection (MOI) of 200. The culture plates were incubated at 37 °C for 2 h, after which the infected cells were washed with PBS. The infected cells were incubated for 1 h in the presence of 50 µg/mL gentamicin to kill extracellular bacteria. The cultures were then placed in fresh DMEM containing 2% FBS and 25 µg/mL gentamicin and incubated at 37 °C under 5% CO_2_.

### 2.6. Determining the Intracellular Survival of Brucella Abortus ΔBspF

HeLa cells were cultured in 6-well plates until the cell count reached 1.0 × 10^6^ cells per well. *Brucella abortus* 2308 and Δ*bspF* mutant strains were infected with a MOI = 1:200. Lysozyme (0.2%) was added at 8, 12, 24, and 48 h after infection to release the bacteria in the cells, diluted to concentrations of 10^−2^, 10^−3^, 10^−4^, and 10^−5^, and coated with TSA. After incubation at 37 °C for 3 days, the number of bacteria in the culture dish was recorded.

### 2.7. Flow Cytometry Analysis

HeLa cells were cultured in 6-well plates until the cell count reached 1.0 × 10^6^ cells per well. The control group, *Brucella* 2308 group, and Δ*bspF* infection group were set up. The *Brucella* 2308 and Δ*bspF* mutant strains were infected at a multiplicity of infection MOI = 1:50 for 24 and 48 h, and the medium was discarded and washed three times with PBS. Cells were digested with trypsin (without EDTA) (Hyclone, Logan, UT, USA) and treated according to the operating instructions of the Annexin V-FITC apoptosis detection kit (Beyotime, Shanghai, China). The samples were then examined using BD FACS Aria III (BD, Franklin Lakes, NJ, USA) flow cytometry. The individual test was repeated three times. The data analysis was performed using FlowJo-10.8.1 software.

### 2.8. Statistical Analysis

All data analysis methods in this study were analyzed by a two-sample equal variance test, and a *p* > 0.05 was considered not significantly different. A *p <* 0.05 indicates that the results of the test data are significantly different. * *p <* 0.05, ** *p <* 0.01, *** *p <* 0.001, **** *p <* 0.0001. The density of the immunoblot band was quantified using ImageJ 1 software. The data graph was generated by GraphPad Prism 6, and each experiment was performed three times.

## 3. Results

### 3.1. Brucella Effector Protein BspF Attenuates Crotonylation of p53

In our previous publication, we found that BspF can attenuate the crotonylation modification of p53 at the K351 site by high-resolution mass spectrometry (MS)-based crotonylation modification proteomics [26]. In order to explore the relationship of p53 and BspF, pCMV-HA-BspF and Flag-tag2B-p53 were constructed successfully and were co-transfected into HEK-293T cells for 30 h, at which the highest BspF protein expression was observed. Anti-Kcr mAb, anti-HA-tag mAb, and anti-Flag-tag mAb were used as the primary antibody, and goat anti-mouse IgG (H + L) was used as the secondary antibody to detect the crotonylation modification of p53 and the protein expression of BspF and p53. BspF contains 428 amino acids and has a size of 48.7 kDa. The p53 protein has a molecular weight of 43.7 kDa; however, its protein band appears at 53 kDa and is consequently named p53. As shown in Figure 1A, the anti-HA and anti-Flag bands in WCL were located at 50 kDa–35 kDa and 60 kDa–48 kDa, respectively, which were consistent with the size of the BspF and p53 protein bands, indicating that BspF and p53 were expressed in the HEK-293T cells. Moreover, the bands of the fourth channel of anti-Kcr were shallower than those of the third channel, indicating that the original crotonylation modification of p53 was weakened after adding BspF, and the overexpression of BspF downregulated the crotonylation modification of p53. Additionally, we also found that BspF interacted with p53 (Figure 1A,B).

We found that BspF attenuated the crotonylation modification of p53. To further verify whether this process is affected by BspF weakening the crotonylation of p53 at the k351 site, we co-transfected BspF with wild-type p53 (Flag-p53) and mutant Flag-tag2B-p53K351A. As shown in Figure 1C, anti-Kcr in the IP indicated the crotonylation modification level, and the second channel had a shallower band than the first channel, indicating that the crotonylation modification of p53 was weakened after adding BspF. The bands of the third and fourth channels were almost unchanged, indicating that the crotonylation modification of p53K351A was almost unchanged after adding BspF. Therefore, Figure 1C shows that BspF affected the crotonylation modification of p53 but had less effect on the crotonylation modification of p53K351A, which indicated that BspF affected the crotonylation modification of p53 by weakening the crotonylation modification of lysine at position 351 of p53.

Subsequently, to determine whether BspF affects the crotonylation of p53 depending on the GNAT domain of BspF, the BspF and BspF GNAT domain deletion (BspFΔGNAT, 34.8 kDa) expression plasmids were co-transfected with p53, respectively. As shown in Figure 1D, the crotonylation of p53 was significantly weakened after the addition of BspF, while the crotonylation of p53 was not significantly weakened after the addition of BspFΔGNAT. Therefore, BspF could affect the crotonylation modification of p53 through the GNAT domain.

### 3.2. BspF Reduces the Expression of p53 Protein

It has been reported that p53 serine crotonylation affects its expression [27]; therefore, we speculated that the lysine crotonylation of p53 could regulate its expression. To explore whether BspF has an effect on p53 protein expression by regulating p53 crotonylation modification, increasing amounts of HA-BspF were transfected with equal doses of wild-type p53 and p53K351A mutant, respectively. As shown in Figure 2A,C, the expression of p53 protein decreased with the increasing quantities of BspF, while the expression of p53K351A protein remained basically unchanged. In addition, the crotonylation of p53 was weakened, and the crotonylation of p53K351A had little effect, which was consistent with the results in Figure 1C.

As shown in Figure 2B, when different doses of BspFΔGNAT were co-transfected with an equal amount of wild-type p53, it was found that an increased BspFΔGNAT dose had no effect on the expression of p53 protein and the crotonylation modification of p53, which was consistent with the results in Figure 1D. These results suggested that BspF regulates the expression of p53 by affecting the crotonylation modification at the K351 site of p53 and is dependent on the GNAT domain.

### 3.3. Brucella ΔbspF Mutant Strain Effects on Cell Apoptosis

BspF can affect the expression level of p53, which is a vital protein in apoptosis signaling pathways. Therefore, we further investigated the effect of BspF on cell apoptosis. We infected HeLa cells with the 2308 WT strain and 2308 Δ*bspF* strain, respectively, and detected apoptosis by flow cytometry at 24 and 48 h. When the cells were infected with Δ*bspF* for 24 h and 48 h, the average apoptosis of the HeLa cells (early apoptosis cells) significantly increased, reaching approximately 4.36 ± 0.13% and 15.66 ± 0.53%, respectively (Table 5 and Table 6). However, the average apoptosis of early apoptosis cells in the 2308 WT strain was approximately 1.76 ± 0.43% and 4.86 ± 0.23%, respectively (Table 5 and Table 6). As shown in Figure 3A,B, compared to the uninfected cells, the apoptosis of 2308 WT cells was intensified after infection, while there was no significant difference in the apoptosis of 2308 Δ*bspF*-infected cells and uninfected cells. These results indicated that the 2308 WT strain inhibits apoptosis, and the deletion of the effector protein BspF strain restores apoptosis to normal levels. BspF has a significant inhibitory effect on host cell apoptosis.

The p53 protein regulates apoptosis by activating its downstream genes, inducing the transcription of pro-apoptotic genes, and inhibiting the transcription of anti-apoptotic genes. Therefore, to confirm the effect of the *Brucella* effector protein BspF on apoptosis, we detected the mRNA transcript levels of the downstream genes of p53. HeLa cells were infected with the WT strain and 2308 Δ*bspF* strain simultaneously. RNA was extracted and reverse-transcribed into cDNA at different times after infection. The transcription of the apoptotic genes p53, Caspase-3, Bcl-2, Bax, Bad, and AIF were detected by real-time PCR (RT-PCR) (Figure 3C). The Caspase-3, p53 and Bcl-2 transcription levels were significantly increased at 3 h in the Δ*bspF*-infected cells (*p <* 0.01) compared to the 2308-infected cells. At 12 h after infection, the transcription levels of AIF, Caspase-3, Bax, p53, Bad, and Bcl-2 were noticeably enhanced, especially for Bad (*p <* 0.01). Similarly, the transcription levels of AIF, Caspase-3, Bax, p53, Bad, and Bcl-2 were significantly increased at 24 h and 48 h after infection, especially Caspase-3, Bax, and Bcl-2 (*p <* 0.01). The results showed that the Δ*bspF* strain promotes the mRNA levels of apoptosis-related gene expression.

Moreover, we verified the effect of BspF on the mitochondrial apoptosis pathway by Western blotting experiments. HeLa cells were infected with the 2308 WT strain and 2308 Δ*bspF* strain for 24 h, and then the cells were collected for Western blotting. As shown in Figure 3D,E, the protein expression levels of cleaved-caspase-3, p53, Bax, and AIF in the 2308 Δ*bspF*-infected cells were significantly higher than those in the 2308 WT-infected cells and the uninfected (NI) cells. The results showed that the Δ*bspF* strain could increase the expression of apoptotic proteins.

### 3.4. BspF Inhibits the Transcription and Protein Expression of Apoptosis-Related Genes through Its GNAT Domain

Since the previous results showed that BspF affects p53 protein expression dependent on the GNAT domain, we want to further explore whether BspF regulates the mitochondrial apoptosis pathway through its GNAT domain. For this purpose, HeLa cells were transfected with BspF or BspFΔGNAT recombinant plasmids, and an equivalent amount of the control vector was also transfected. We found that BspF inhibited the mRNA transcription of p53, Caspase-3, Bcl-2, Bax, and AIF, while BspFΔGNAT had no significant effect (Figure 4A), indicating that BspF relies on the GNAT domain to inhibit the mRNA transcription levels of mitochondrial apoptosis-related genes.

Furthermore, to investigate at the protein level whether BspF affects the expression of apoptosis-related proteins through its GNAT domain, different doses of BspF and BspFΔGNAT were transfected into HeLa cells, respectively. As shown in Figure 4B, the protein expression of p53, Caspase-3, AIF, and Bax decreased with the increiase of BspF expression, while the increasing dose of BspFΔGNAT had no effect on the expression of these proteins. The results suggested that BspF is involved in *Brucella*-mediated apoptosis through the mitochondrial apoptosis pathway (MAP) by the GNAT domain.

### 3.5. BspF Can Promote the Survival of Brucella

To determine whether the deletion of BspF affects the intracellular survival of *Brucella*, the number of CFUs was counted following infection, with *Brucella abortus* 2308 and 2308 Δ*bspF*. As shown in Figure 5, after 8 h and 12 h of infection, there was no significant change in the intracellular survival of 2308 and 2308 Δ*bspF*. At 24 h post-infection, the intracellular survival amount of the Δ*bspF* strain was lower than 2308WT, and the intracellular viability of the Δ*bspF* strain was significantly reduced (*p <* 0.05) at 48 h. The results suggested that a BspF deficiency inhibits the intracellular survival of *Brucella* and may act as an effector protein, playing an important potential virulence role.

## 4. Discussion

Crotonylation of lysine (Kcr) is a new histone modification that was first identified in 2011 by Tan et al., who used mass spectrometry to identify 67 new post-translational modification (PTM) sites in HeLa cells and mouse spermatogonia cells [28]. Crotonylation can shield the positive charge of histones, making the binding of negatively charged DNA and histones looser, which is conducive to the binding of transcription factors and DNA to promote the transcription process [29]. Non-histone croton acylation modification is the discovery of 558 specific croton acylation modification sites in HeLa cells in 2017, indicating that croton acylation modification can regulate multiple protein functions and cell growth processes [30]. The transcriptional co-activator, p300, has both histone acetyltransferase (HAT) activity and histone crotonyltransferase (HCT) activity, and p300-catalyzed histone crotonylation directly stimulates transcription to a greater degree than does p300-catalyzed histone acetylation [31]. GNAT is one of the most powerful and widely distributed acetyltransferase families, which many acetyltransferases also have crotonyltransferase activity, and BspF has been found to have de-crotonyltransferase activity in our previous studies [26].

*Brucella* is a typically intracellular parasite that survives in host cells through endotoxin, T4SS, and cytochrome [32]. T4SS is one of the main virulence factors, which is indispensable in regulating the intracellular replication of *Brucella* and is important to host-persistent infection [19,20,32]. T4SS promotes the secretion of effector proteins, thereby inhibiting the fusion of BCV and lysosomes and promoting the transfer of *Brucella* to the endoplasmic reticulum, where it creates replication sites [21]. Studies have shown that *Brucella* invades host cells, and the effector proteins BspA, BspB, and BspF, secreted by T4SS, can reduce the secretion of host proteins, thus escaping immune recognition and promoting *Brucella’s* sustained survival in host cells. In order to find the target protein of BspF acting on host cells, important proteins in the immune pathway were selected from the crotonylation omics data, and it was found that BspF affected the crotonylation modification at the K351 site of p53.

As a pro-apoptotic factor, p53 participates in apoptosis by transcribing important genes in the mitochondrial apoptosis pathway and the death receptor apoptosis pathway [33]. In the mitochondrial apoptotic pathway, p53 mainly activates the expression of downstream genes, such as Bax, PUMA, Noxa, and Bi, which increases the mitochondrial membrane permeability and affects the release of cytochrome C and ATP in the cytoplasm [34]. Furthermore, p53 binding to Apaf-l forms an apoptotic precursor to activate Caspase-9 and cleaves the protease Caspase-3, which can cause cascade reactions and lead to apoptosis [35]. In our results (Figure 1 and Figure 2), we identified that the effector protein BspF of *Brucella* can interact with p53 protein. We also verified that BspF attenuates the crotonylation modification of p53 at K351 via its GNAT domain, thereby affecting the overall crotonylation modification of p53. Moreover, we found that BspF can inhibit the expression of p53 protein, and this process depends on the effect of the GNAT domain on the p53K351 site. From the above experimental results, it is concluded that BspF depends on the GNAT domain to attenuate the crotonylation modification of p53, thereby reducing its expression. We speculate that one possibility is that the de-crotonyltransferase activity of BspF directly affects the crotonylation modification of p53 by acting on p53K351, and the other possibility is that BspF indirectly affects the crotonylation modification of p53 by interacting with other transferase proteins or by blocking the K351 crotonylation site.

When the pathogen invades the host cell, it will directly or indirectly affect the normal physiological function of the host cells. The persistent infection of the pathogen in the host is inseparable from its interaction with the apoptotic pathway [36]. Unlike many pathogens released after the death of host cells, *Brucella* can prevent the death of host cells from maintaining their intracellular living environment [37]. Manipulating host cell death is a key strategy for *Brucella* in maintaining transmission and intracellular survival. The effector protein VceC promotes persistent *Brucella* intracellular infection by interacting with Grp78, reducing Grp78 expression, and inhibiting CHOP-induced apoptosis. BspJ is a nuclear regulatory protein secreted by *Brucella*, and a study has reported that the BspJ deletion strain promotes apoptosis after cell infestation and reduces the survival of *Brucella* within macrophages [38]. Therefore, the *Brucella* BspJ protein can also inhibit cell apoptosis. These studies have suggested that *Brucella* effector proteins play a vital role in regulating cell apoptosis. These studies have suggested that *Brucella* can influence apoptosis through its effector proteins. The results of Figure 2 showed that BspF affected the crotonylation modification of p53 through its GNAT domain, thus affecting the expression of p53. As an important marker protein of the mitochondrial apoptosis pathway, p53 affects the transcription level of downstream genes [39]. It indicates that the effector protein BspF may have an effect on the apoptosis pathway; therefore, we further explored the effect of BspF on apoptosis.

Caspase-3 is a key executor in apoptosis, and apoptosis is more pronounced when the expression level of Caspase-3 is improved [40]. Compared with *Brucella* 2308, the Δ*bspF* strain infection significantly increased the mRNA transcription level of Caspase-3 at 3, 12, 24, and 48 h after infection, and it also significantly increased the expression of cleaved-caspase-3 protein. As an apoptosis-inducing factor, AIF interacts with mitochondrial protein endonuclease G (EndoG) to cause the fragmentation of condensed DNA of the cell chromosomes and apoptosis of cells [41]. Compared with *Brucella* 2308, the transcription level of the AIF mRNA gene was significantly increased at 12, 24, and 48 h in 2308 Δ*bspF*-infected cells, and the expression of AIF protein was also significantly increased. As a pro-apoptotic protein, Bax is the main mediator of the mitochondrial apoptosis pathway. Compared with *Brucella* 2308, the mRNA transcription level of Bax was significantly increased at 12, 24, and 48 h after *Brucella* Δ*bspF* infection (Figure 3C), and the protein expression of Bax was also significantly increased (Figure 3D,E). These results indicated that BspF inhibits apoptosis. The data shown in the results of flow cytometry further support the notion that BspF assumes a critical role in inhibiting apoptosis, which demonstrated that the apoptosis of the cells infected with the *Brucella* Δ*bspF* strain was significantly higher after 24 and 48 h than the 2308 WT strain (Figure 3A,B).

In addition, by detecting the mRNA transcription level and protein expression level of apoptosis-related factors, we found that BspF relies on the decrotonylase activity of the GNAT domain to inhibit apoptosis. At the same time, it was also found that the intracellular viability of the *Brucella* Δ*bspF* strain was significantly reduced compared with the 2308 WT strain, indicating that the deletion of the effector protein BspF would affect the intracellular survival of *Brucella*. One of the strategies for *Brucella* to invade the host is to protect the cells and inhibit cell apoptosis so that *Brucella* can exist stably in BCVs and support *Brucella* in surviving, reproducing, and persistently infecting the host [38]. We hypothesize that the effector protein BspF could change the crotonylation modification of some proteins, affecting their functions and signaling pathways in the cell, thus providing suitable conditions for *Brucella* growth and reproduction. To understand the mechanisms by which *Brucella* affects cell apoptosis and survives in the host for extended periods, it is important to investigate how other intracellular bacteria infect the host. Taken together, our findings reveal that the BspF plays an important role in regulating host cell processes, promoting the intracellular survival of *Brucella* and triggering the apoptosis of host cells. The findings constitute an important reference for the study of non-histone crotonylation modification and also provide a reference for *Brucella* evading the immune response of host cells.

## 5. Conclusions

This study provides valuable information regarding *Brucella’s* T4SS effector proteins affecting host cell apoptosis and escaping the host immune response. According to the proteomics data, we discovered that the *Brucella* effector protein BspF can regulate the crotonylation modification of p53 in host cells through its GNAT domain and support the persistent survival of *Brucella* by inhibiting cell apoptosis. These findings provided new insights into exploring the immune response mechanism of the host in response to intracellular bacterial infection.

## Figures and Tables

**Figure 1 microorganisms-11-02322-f001:**
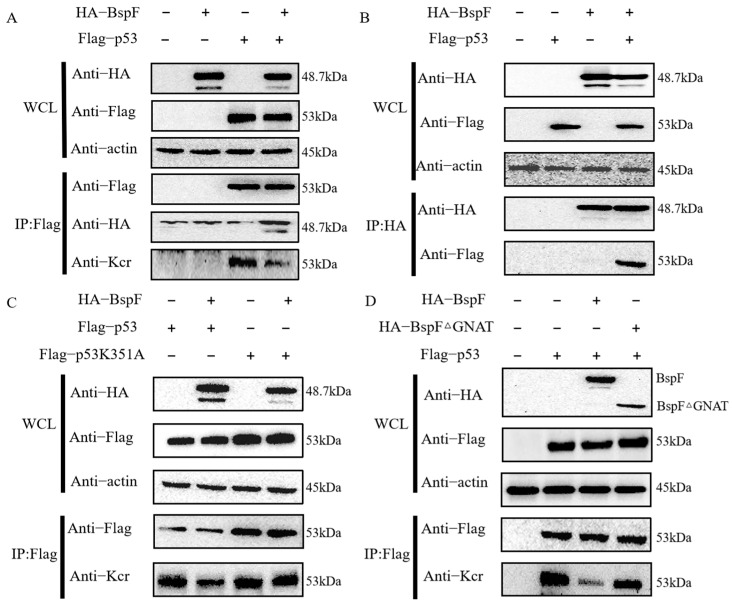
BspF attenuates the crotonylation modification of p53 by the GNAT domain and also interacts with p53. (**A**) The effect of BspF on the crotonylation of p53 and the interaction between BspF and p53 was detected by co-immunoprecipitation with anti-Flag-tag antibody; (**B**) with HA-tagged antibody, and WCL represents whole cell lysate; (**C**) with FLAG-tagged antibody. (**D**) The effects of BspF and BspFΔGNAT on the crotonylation of p53 were detected by co-immunoprecipitation.

**Figure 2 microorganisms-11-02322-f002:**
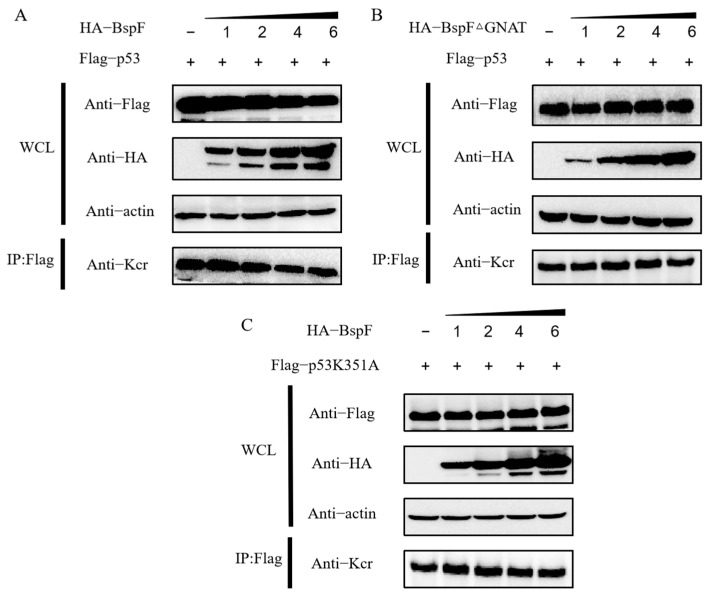
BspF affects p53K351 site crotonylation modification through its GNAT domain and affects p53 protein expression. (**A**) The effects of increasing BspF transfection dose on p53 protein expression and crotonylation modification were detected by Western blotting. (**B**) Western blotting was used to detect the effect of the increased transfection dose of BspFΔGNAT on p53 protein expression and crotonylation modification. (**C**) The effects of increasing BspF transfection dose on p53K351A protein expression and crotonylation modification were detected.

**Figure 3 microorganisms-11-02322-f003:**
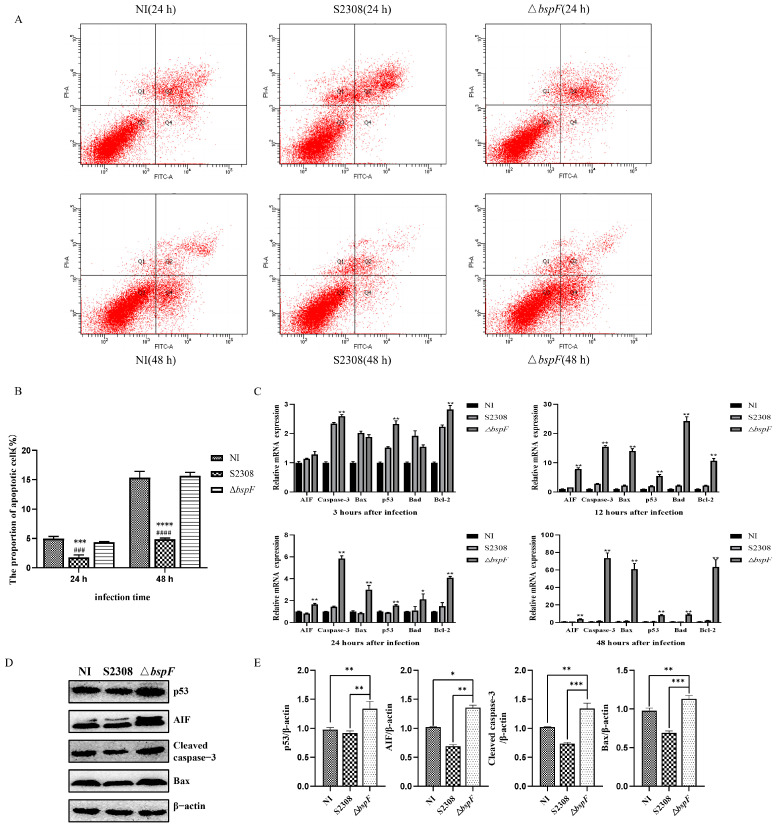
*Brucella* Δ*bspF* strain promoted the transcription of apoptosis-related genes and protein expression and affected cell apoptosis. (**A**) Flow cytometry was used to detect cell apoptosis of different treatment groups. Cells were stained with propyridine iodide (PI) and Annexin V-FITC. Cells with Annexin V-FITC-positive and PI-negative staining were apoptotic cells (lower right quadrant). (**B**) The fraction of apoptotic cells. * indicates that the results of the experimental data in this group are significantly different from those in NI group. * *p* < 0.05, ** *p* < 0.01, *** *p* < 0.001, **** *p* < 0.0001. # indicates that the results of the experimental data in this group are significantly different from those in ΔbspF group. ### *p* < 0.001, #### *p* < 0.0001. (**C**) mRNA expression of apoptosis-related genes AIF, Caspase-3, p53, Bax, Bcl-2, and Bad relative to β-actin in each group at each time point. ** Indicates the significant difference between this group and the 2308 group (*p <* 0.01), * Indicates the significant difference between this group and the 2308 group (*p <* 0.05). (**D**) Immunoblotting of p53, AIF, Bax, and cleaved-caspase-3 proteins in HeLa cells not infected or infected with different *Brucella* 2308 strains. (**E**) Ratio of the apoptotic marker proteins p53, AIF, cleaved-caspase-3, and Bax to β-actin protein. The density of the immunoblot band was quantified using ImageJ 1 software. The strip density values of each p53, AIF, cleaved-caspase-3, and Bax were divided by the corresponding strip density values of β-actin for each channel and repeated three times.

**Figure 4 microorganisms-11-02322-f004:**
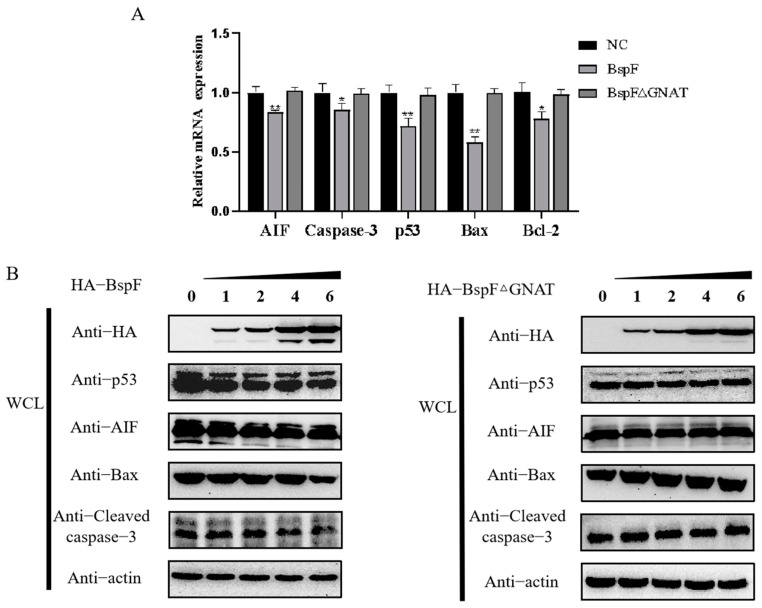
BspF inhibits the transcription and protein expression of apoptosis-related genes through its GNAT domain. (**A**) mRNA expression of apoptosis-related genes AIF, Caspase-3, p53, Bax, and Bcl-2 relative to β-actin. Each group had three replications. ** Indicates the significant difference between this group and the BspFΔGNAT group (*p <* 0.01); * indicates the significant difference between this group and the BspFΔGNAT group (*p <* 0.05). (**B**) In HeLa cells transfected with increasing doses of HA-BspF and HA-BspFΔGNAT, endogenous antibodies were used to detect changes in the levels of apoptotic markers.

**Figure 5 microorganisms-11-02322-f005:**
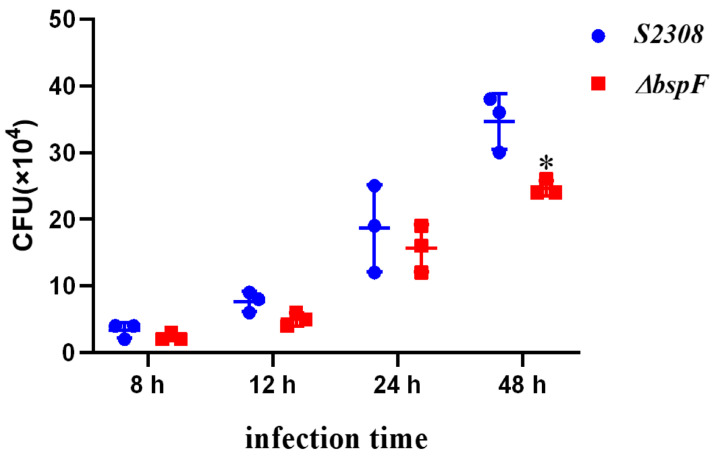
BspF enhances the intracellular survival of *Brucella*—the viability of *Brucella abortus* 2308 and its Δ*bspF* strain in HeLa cells. HeLa cells were infected with the 2308 wild-type strain and Δ*bspF* strain at 8, 12, 24, and 48 h after infection. Each experiment was performed three times. CFU was counted. * Indicates a significant difference between this group and the 2308 group (*p <* 0.05).

**Table 1 microorganisms-11-02322-t001:** The reaction system of PCR.

Reactive Component	Volume (µL)
cDNA	1
Forward primer	1
Reverse primer	1
2× Prime STAR Max Premix	10
ddH_2_O	7
Total volume	20

**Table 2 microorganisms-11-02322-t002:** The primer sequences for gene amplification of p53.

Primer Name		Primer Sequences (5′-3′)	Target Fragment (bp)	Restriction Endonucleases
Flag-p53	Forward:	AAAGAATTCATGGAGGAGCCGC	1182	*EcoR* I
	Reverse:	AAACTCGAGTCAGTCTGAGTCAGGC		*Xho* I
Flag-p53K351A	Forward:	AGGCCTTGGAACTCGCGGATGCCCAGGCTGG	1182	*EcoR* I
	Reverse:	CCAGCCTGGGCATCCGCGAGTTCCAAGGCCT		*Xho* I

**Table 3 microorganisms-11-02322-t003:** Fluorescence quantitative PCR reaction system.

Reactive Component	Volume (µL)
cDNA	1
Forward primer	0.5
Reverse primer	0.5
SYBR qPCR Master Mix	5
ddH_2_O	3
Total volume	10

**Table 4 microorganisms-11-02322-t004:** The primer sequences.

Primer Names		Primer Sequence (5′-3′)	Product Length (bp)
p53	Forward:	ATGAGCCGCCTGAGGTTGG	71
	Reverse:	CAGTGTGATGATGGTGAGGATGG	
Caspase-3	Forward:	GTGGAATTGATGCGTGATG	193
	Reverse:	TCTCAATGCCACAGTCCAGT	
AIF	Forward:	CGGCTCCCAGGCAACTTGTTC	104
	Reverse:	GGCACCAGCTCCTACTGTTGATAAG	
Bcl-2	Forward:	GGCTACGAGTGGGATGCG	168
	Reverse:	TGGACCACAGGTGGCAC	
Bax	Forward:	AGAGGATGATTGCCGCC	117
	Reverse:	GTGCACAGGGCCTTGAG	
Bad	Forward:	CGGAGGATGAGTGACGAGTTTGTG	79
	Reverse:	GATCCCACCAGGACTGGAAGACTC	
β-actin	Forward:	AAAGACCTGTACGCCAACAC	178
	Reverse:	GTCATACTCCTGCTTGCTGAT	

**Table 5 microorganisms-11-02322-t005:** Results of Annexin V-FITC/PI staining for cell apoptosis after *B. abortus* at 24 h.

Group	Normal Cells	The Early Apoptotic Cells (%)	The Late Apoptotic Cells (%)
Control (NI)	84.70 ± 0.39	4.96 ± 0.43	11.94 ± 1.45
S2308-infected	83.40 ± 0.54	1.76 ± 0.43	17.93 ± 1.33
Δ*bspF*-infected	83.40 ± 0.26	4.36 ± 0.13	14.60 ± 0.46

**Table 6 microorganisms-11-02322-t006:** Results of Annexin V-FITC/PI staining for cell apoptosis after *B. abortus* at 48 h.

Group	Normal Cells	The Early Apoptotic Cells (%)	The Late Apoptotic Cells (%)
Control (NI)	73.70 ± 0.37	15.36 ± 1.13	5.7 ± 1.10
S2308-infected	75.40 ± 0.45	4.86 ± 0.23	4.93 ± 0.63
Δ*bspF*-infected	74.40 ± 0.74	15.66 ± 0.53	5.9 ± 0.23

## Data Availability

The data presented in this study are available upon request from the corresponding author.

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
