# Peer review of "The Brucella Effector Protein BspF Regulates Apoptosis through the Crotonylation of p53"

_microorganisms, 2023, doi:10.3390/microorganisms11092322_

Round 1

Reviewer 1 Report

Major points:

L28: Abstract and main manuscript state that BspF decrotonylates p53. However, they authors did not show any experiment that directly demonstrates this mechanism. It is only shown that WT BspF affects p53 K351 crotonylation, but this may also be achieved by an indirect mechanism or by blocking the K351 crotonylation site. In none of the experiments an enzymatic assay was done to prove a deamidase activity of BspF.

Instead, the authors must clearly state that their experiments only show that BspF negatively regulates p53 crotonylation and that they speculate BspF hydrolyses the acylamide at K351  based on previous experiments done with histones published by the authors in doi: 10.3389/fcimb.2020.603457.

The majority of methods lack sufficient detail for reproducibility, and sometimes even proper understanding. Carefully verify, where methods require additional description. I only give examples, there is just too much missing here.

Example 1:

Line 126: cell culture is not described, no infos about cultivation and transfection

Example 2:

Line 134 Co-IP lacks essential details, e.g. how much supernatant was used, its protein concentration, how ECL signal was detected (imaging system?). Furthermore, I doubt you used ECL system for color development – it should be used in combination with imaging system to record the produced chemiluminescence.

Example 3:

Line 172: Flow cytometry no instrument settings are provided, what are well-grown HELA cells, how many cells were used?

L207: you claim that results demonstrate specificity for p53K351 only – how can you say this, given that no other crotonylation sites were investigated? Remove “specifically”, this statement is misleading.

Your experimental results depicted in Fig1 does not fit to your interpretation; in fact, there are results that are questionable or need further explanation:

Indicate the MW of WB bands and write in text, if they fit to expected MW or not.

Anti-HA antibody displays frequently two (A,B,C) or even three bands (D) in WCL. Moreover, even without expression of full-length HA-BspF, at least one band remains (D). Which band in IP corresponds to band in WCL?

You state that BspF overexpression decreases p53 crotonylation. Without a quantitative analysis (similar to Fig2) this result is questionable. Either provide additional data (more experimental replicates, dilution series) or state that your results suggest decreased crotonylation, but additional work is required to confirm this.

Fig1D has serious problem. There is no signal present for BspF-delta-GNAT on the WCL Western you show. I seriously doubt it is the band with lowest MW, because deletion of such big domain should result in significantly different migration in SDS-PAGE between full length and delta-GNAT BspF. Hence, it does not confirm any effect of GNAT domain on crotonylation of p53, too.

Minor points:

L83: sentence is unclear, VceC is effector protein and first identified substrate of Brucella – why is it a Brucella substrate and not effector?

L190: Here we reported… statement is unclear – to my understanding you talk about previous publication and not current one, revise sentence.

It should be discussed, how similar the concentration of BspF from HeLa cell transfection is to a cell infected with Brucella in order to estimate how closely your model resembles real infection.

Fig1: anti-FLA-tag antibody is what you use, not Flag-tagged

L244: deletion of BspF does not promote apoptosis, but it returns to normal (without infection), revise statement.

Fig3 A: what does the y-axis label mean? Explain.

Fig3 B: you write apoptosis rate, but rate is linked to kinetics (e.g. apoptosis/time). This is confusing. I recommend using another word, e.g. fraction of apoptotic cells.

Fig 3E: how the ratio was calculated/ the WB quantified and how many replicates were used must be explained

Fig 4A: how many replicates?

Fig5: how many replicates? Why 5A, since there is no 5B?

L318: cite reference(s) for this statement.

L320: …is closely related to… , do you mean “is important  for”? Rephrase for clarity

L342: I suggest you present possible hypotheses for the mechanism underlying attenuated crotonylation

English:

Some sentences need revision for clarity, and there are some grammatical mistakes in the manuscript, for example:

L42/43 contact with

L46 BCVs further induce

L49 associated with

L206 affection -> you mean effect?

Author Response

Major points:

  1. L28: Abstract and main manuscript state that BspF decrotonylates p53. However, they authors did not show any experiment that directly demonstrates this mechanism. It is only shown that WT BspF affects p53 K351 crotonylation, but this may also be achieved by an indirect mechanism or by blocking the K351 crotonylation site. In none of the experiments an enzymatic assay was done to prove a deamidase activity of BspF.Instead, the authors must clearly state that their experiments only show that BspF negatively regulates p53 crotonylation and that they speculate BspF hydrolyses the acylamide at K351  based on previous experiments done with histones published by the authors in doi: 10.3389/fcimb.2020.603457.

Reply: We sincerely thank the reviewer for this very constructive suggestion. Our description in the text indeed only show that WT BspF affects p53 K351 crotonylation. Following your suggestions, we speculate that one possibility is that the decrotonyltransferase activity of BspF directly affects the crotonylation modification of p53 by acting on p53K351, and the other possibility is that BspF indirectly affects the crotonylation modification of p53 by interacting with other transferase proteins or by blocking the K351 crotonylation site and added it in the revised manuscript. (Line467-470) We have cited doi: 10.3389/fcimb.2020.603457 in the revised manuscript, and have clearly stated in the revised manuscript that BspF negatively regulates p53 crotonylation.

The majority of methods lack sufficient detail for reproducibility, and sometimes even proper understanding. Carefully verify, where methods require additional description. I only give examples, there is just too much missing here.

Example 1:

  1. Line 126: cell culture is not described, no infos about cultivation and transfection

Reply: Thank the reviewer for pointing out this omission. We have added details about cell culture and transfection in the revised manuscript. HEK-293T cells were cultured in a 10cm cell culture dish with a cell density of 5×107 cells per well. The cell culture dish was placed in a cell incubator set at 37 ℃ and 5% CO2. (Line 142-144). When the cell density reached 80∼90%, Lipofectamine 2000 (VigoFect, China) was used to transfect the plasmids (pCMV-HA-BspF, Flag-tag2B-p53) into the HEK-293T cells. (Line146-147)

Example 2:

  1. Line 134 Co-IP lacks essential details, e.g. how much supernatant was used, its protein concentration, how ECL signal was detected (imaging system?). Furthermore, I doubt you used ECL system for color development – it should be used in combination with imaging system to record the produced chemiluminescence.

Reply:We sincerely apologize for our omission. We used 1000µL of supernatant and 40µL Protein A+G agarose beads (Beyotime, China) for CO-IP test. According to the manufacturer's instructions, we used the gel imager (aplegen, USA) to detect the signal intensity of western blotting band with the ultrasensitive ECL chemiluminescence solution (epizyme, China). In addition, we added these details in the revised manuscript. (Line 157-175)

Example 3:

  1. Line 172: Flow cytometry no instrument settings are provided, what are well-grown HELA cells, how many cells were used?

Reply:Thank the reviewer for pointing out this omission. We have added more detailed steps in the revised manuscript: HeLa cells were cultured in 6-well plates and that the number reached 1.0 × 106 cells/well. The cells were evenly spread in six-well plates, and the control group, Brucella 2308 group and ΔbspF infection group were respectively set up. Brucella 2308 and ΔbspF mutant strains were infected at multiplicity of infection MOI= 1:50 for 24 and 48 h, and the medium was discarded and washed three times with PBS. The cells were digested with trypsin (without EDTA) (Hyclone, USA) and treated according to the operating instructions of Annexin V-FITC apoptosis detection kit (Beyotime, China) operating instructions. The samples were then examined using BD FACS Aria III (BD, USA) flow cytometry. Each sample was repeated three times. Data analysis was performed using the FlowJo software. (Line 199-207) (Line 200)

  1. L207: you claim that results demonstrate specificity for p53K351 only – how can you say this, given that no other crotonylation sites were investigated? Remove “specifically”, this statement is misleading.

Reply:We apologized for incorrect description, and we have deleted “specifically”in the revised manuscript. (Line 276)

  1. Your experimental results depicted in Fig1 does not fit to your interpretation; in fact, there are results that are questionable or need further explanation. Indicate the MW of WB bands and write in text, if they fit to expected MW or not.

Reply:We are sorry that the results are illegible. Therefore, we have added the following contents to the revised manuscript: BspF contains 428 amino acids and has a size of 48.7 kDa. p53 protein has a molecular weight of 43.7 kDa, but its protein band appears at 53 kDa, so it is named p53. (Line250-255)

  1. Anti-HA antibody displays frequently two (A, B, C) or even three bands (D) in WCL. Moreover, even without expression of full-length HA-BspF, at least one band remains (D). Which band in IP corresponds to band in WCL?

Reply:We sincerely thank reviewer for the kind remarks. The top bands of figure1 of Anti-HA are the band corresponding to the WCL in the co-IP. We have labelled it in Figure 1A in the revised manuscript.

  1. You state that BspF overexpression decreases p53 crotonylation. Without a quantitative analysis (similar to Fig2) this result is questionable. Either provide additional data (more experimental replicates, dilution series) or state that your results suggest decreased crotonylation, but additional work is required to confirm this.

Reply:Based on your comments, we repeated the experiment to verify that p53 protein crotonylation decreased with increasing doses of BspF and that BspF lacking the GNAT structural domain had no effect on p53 protein crotonylation, and no change in p53 crotonylation modification when the K351 site was mutated. Images were replaced and descriptions added. (Line 305-307) In addition, we've replaced Figure 2 in the revised manuscript.

  1. Fig1D has serious problem. There is no signal present for BspF-delta-GNAT on the WCL Western you show. I seriously doubt it is the band with lowest MW, because deletion of such big domain should result in significantly different migration in SDS-PAGE between full length and delta-GNAT BspF. Hence, it does not confirm any effect of GNAT domain on crotonylation of p53, too.

Reply:We are sincerely sorry for our omission. Due to our mistake, we cut out the GNAT strips when we saved the image. We performed a new assay using the newly prepared sample. As you shown in the figure  stated , the increase of  BspFΔGNAT dose had no effect on p53 protein expression (Figure 2B),and we have substituted Figure 1D in the revised manuscript.

Minor points:

  1. L83: sentence is unclear, VceC is effector protein and first identified substrate of Brucella – why is it a Brucella substrate and not effector?

Reply:We apologize for the error. We have changed “identified substrates” to “effector”. (Line87)

  1. L190: Here we reported… statement is unclear – to my understanding you talk about previous publication and not current one, revise sentence.

Reply:We're really sorry for our negligence, we have changed the sentence to “In our previous publication, we have found that BspF can attenuate the crotonylation modification of p53 at the K351 site by high-resolution mass spectrometry (MS)-based crotonylation modification proteomic” in the revised manuscript. (Line 239~241)

  1. It should be discussed, how similar the concentration of BspF from HeLa cell transfection is to a cell infected with Brucella in order to estimate how closely your model resembles real infection.

Reply:We sincerely appreciate the scientific advice. To investigate the functional roles of BspF in simulating actual infection conditions, we infected wild-type and deletion strains and examined apoptosis indicators in cells which are the transcription levels of AIF, Caspase-3, Bax, p53, Bad, and Bcl-2. At the same time, we overexpressed BspF protein in cells by transfecting the plasmids pCMV-HA-BspF and pCMV-HA-BspFΔGNAT mainly to explore the function of BspF at protein levels,and the results can correspond to the results of bacterial infection. Therefore, we detected apoptotic gene and protein levels in both overexpressed BspF cells and Brucella infected cells.

  1. Fig1: anti-FLA-tag antibody is what you use, not Flag-tagged

Reply:We are sincerely sorry for the mistake. We have change “antibody Flag-tagged” to “anti-Flag-tag”. (Line 290)

  1. L244: deletion of BspF does not promote apoptosis, but it returns to normal (without infection), revise statement.

Reply:We apologize for our incorrect description. We have revised the presentation based on the reviewer's suggestion to “These results indicated that the 2308 WT inhibits apoptosis, and the deletion of effector protein BspF restores apoptosis to normal levels. BspF has significant inhibitory effect on host cell apoptosis”. (Line334-336)

  1. Fig3 A: what does the y-axis label mean? Explain.

Reply:The y-axis is the SSC, lateral corner scattering, and its value represents the cellular granularity. Cell SSC values can be used to compare cell granularity and to group and classify cells.

  1. Fig3 B: you write apoptosis rate, but rate is linked to kinetics (e.g. apoptosis/time). This is confusing. I recommend using another word, e.g. fraction of apoptotic cells.

Reply:Your suggestion is much appreciated, in the note section of Figure 3b in the article, we changed "Bar graph of apoptosis rates (%)" to "The fraction of apoptotic cells (%)". (Line 365)

  1. Fig 3E: how the ratio was calculated/ the WB quantified and how many replicates were used must be explained

Reply:Thank you for your constructive comments. We quantified the immunoblot band density through ImageJ software. The band density values of p53, AIF, Cleaved caspase-3 and Bax in each channel were divided by the corresponding β-actin band density value, and three replicates were used to ensure accuracy. (Line372-375)

  1. Fig 4A: how many replicates?

Reply:We apologize for the lack of description about replication times in the paper. We conducted three replications of each experiment, which have now been included in the revised manuscript. (Line 399)

  1. Fig5: how many replicates? Why 5A, since there is no 5B?

Reply:Thank you very much for your comments. We conducted three replications of each experiment. We apologize for the incorrect appearance of the serial number in the image due to our negligence. We have now corrected it in the revised manuscript. (Line 417)

  1. L318: cite reference(s) for this statement.

Reply:We apologize for our negligence, and we have cited the literature (DOI: 10.1016/j.micpath.2017.03.007) in the revised manuscript. (Line 434)

  1. L320: …is closely related to… , do you mean “is important  for”? Rephrase for clarity.

Reply:Following to the reviewer’s suggestion, we changed "closely related to" to "important to”. (Line 437)

  1. L342: I suggest you present possible hypotheses for the mechanism underlying attenuated crotonylation.

Reply:Following to the reviewer’s suggestion, we put forward some possible hypotheses on the mechanism of attenuated crotonylation in the revised manuscript as follows: “We speculate that one possibility is that the de-crotonyltransferase activity of BspF directly affects the crotonylation modification of p53 by acting on p53K351, and the other possibility is that BspF indirectly affects the crotonylation modification of p53 by interacting with other transferase proteins or by blocking the K351 crotonylation site”. (Line 522-525)

English:

  1. Some sentences need revision for clarity, and there are some grammatical mistakes in the manuscript, for example:

Reply:We sincerely thank reviewer for the kind remarks, and we've revised our grammar throughout.

  1. L42/43 contact with

Reply:Thank the reviewer for pointing out this omission. We changed the “contact with” to “exposure to” in the section. (Line 46)

  1. L46 BCVs further induce.

Reply:Thank the reviewer for pointing out this omission, we have changed "BCVs further induce " to " BCVs further induces ". (Line 55)

  1. L49 associated with

Reply:We would like to thank the reviewer for pointing out our omissions. We added "with" in the revised manuscript. (Line 57)

  1. L206 affection -> you mean effect?

Reply:Following to the reviewer’s suggestion, we changed "affection" to "effect”. (Line 274)

Reviewer 2 Report

1. When you start your introduction with a paragraph full of inaccuracies, you indicate to the reader that you either don't know Brucella or you don't care to offer proper information. So: Brucella abortus is NOT THE causative agent. It is the 2nd most common agent, B. melitensis being the predominant for humans (we are supposedly focusing on humans after all). And there causative agents are rapidly expanding, but even a decade ago, there were more than abortus....Second, what is the Class I infectious agents of WHO? Can you refer to it? Because your reference here is an article about Kenya, that doesn't include such a comment. WHO has brucellosis as one of the neglected zoonoses though. Third, does WHO reproduce the annual 500,000 cases number, as you write? Please refer to where or change. In fact, please read a bit about brucellosis, apart from its cellular interactions (well referenced, these), and write something proper and accurate. 

2. Crotonylation is something new- of the past 12 years, in the literature. It has a central role in the manuscript. It needs introduction, and further analysis, as a mechanism, in the discussion. 

3. Brucella's interaction with p53 as a therapeutic target? You really imply to mess with p53?

Author Response

  1. When you start your introduction with a paragraph full of inaccuracies, you indicate to the reader that you either don't know Brucella or you don't care to offer proper information. So: Brucella abortus is NOT THE causative agent. It is the 2nd most common agent, B. melitensis being the predominant for humans (we are supposedly focusing on humans after all). And there causative agents are rapidly expanding, but even a decade ago, there were more than abortus.... Second, what is the Class I infectious agents of WHO? Can you refer to it? Because your reference here is an article about Kenya, that doesn't include such a comment. WHO has brucellosis as one of the neglected zoonoses though. Third, does WHO reproduce the annual 500,000 cases number, as you write? Please refer to where or change. In fact, please read a bit about brucellosis, apart from its cellular interactions (well referenced, these), and write something proper and accurate.

Reply:We apologize for the misrepresentation. According to your suggestions, in the revised manuscript, we have described Brucella more correctly, and we changed the content to “Brucellosis is a naturally epidemic human-animal disease caused by Brucella. The spectrum of Brucellosis infection is very wide, and Brucellosis in clinical recessive or chronic infection is easy to miss and misdiagnose, resulting in widespread prevalence of the disease in more than 170 countries” (Line 34-39). In addition, in order to make our representation more correct and accurate, we have deleted the reference to the WHO's definition of Category I infectious agents and its reporting of more than 500,000 cases per year. (Line 39-43)

  1. Crotonylation is something new- of the past 12 years, in the literature. It has a central role in the manuscript. It needs introduction, and further analysis, as a mechanism, in the discussion.

Reply:We sincerely thank the reviewer for this constructive suggestion. We have added more detailed and deep going introduction and analysis of the mechanism of crotonylation in the discussion section in the revised manuscript. (Line 420-434)

  1. Brucella's interaction with p53 as a therapeutic target? You really imply to mess with p53?

Reply:We apologize for the inaccurate conjecture regarding that Brucella's interaction with p53 is a therapeutic target. Without experimental validation, it cannot be considered as such. We have deleted “as well as theoretical basis for informing the development of a treatment strategy for Brucella infection”. (Line532~533)

Reviewer 3 Report

The authors explored how Brucella can affect host cell apoptosis and can evade the host immune response. The study provides a reference for investigating the host immune response mechanism against intracellular bacterial infections, serves as an important reference for studying non-histone crotonylation modification and establishes a theoretical basis for the prevention and treatment of brucellosis.

Major issues

The full details (not just the sequences of primers) of all the PCRs performed in the study must be described in relevant tables. This will allow to evaluate the correct identifications.

Presentation of results. The authors must improve the presentation of results in tabular form rathe than in text.

The Discussion can be extended – as it is now, it seems rather shallow. Also, significant relevant references are missing. Some passages from the Introduction can be transferred to the Discussion.

Overall. The manuscript requires improvement before possible acceptance.

Author Response

  1. The full details (not just the sequences of primers) of all the PCRs performed in the study must be described in relevant tables. This will allow to evaluate the correct identifications.

Reply:We sincerely thank the reviewer for this very constructive suggestion. In the revised manuscript, we have provided the detailed description of the PCR reaction system (Table 1 and 2) and the instrument interface, which includes: predenaturation at 94 ℃ for 5 min, denatured at 94 ℃ for 30 s, annealed at 56 ℃ for 30 s, extended at 72 ℃ for 30 s, extended at 72 ℃ for 10 min for 30 cycles, and preserved at 4 ℃ after the reaction. (Line 124-127)

Table 1. The reaction system of PCR

Reactive component

Volume (µL)

cDNA

1

Forward primer

1

Reverse primer

1

2× Prime STAR Max Premix

10

ddH2O

7

Total volume

20

Table 2. Fluorescence quantitative PCR reaction system

Reactive component

Volume (µL)

cDNA

1

Forward primer

0.5

Reverse primer

0.5

SYBR qPCR Master Mix

5

ddH2O

3

Total volume

10

  1. Presentation of results. The authors must improve the presentation of results in tabular form rather than in text.

Reply:Thank you for your constructive comments. According to the reviewer’s suggestion, we have added several new tables in the revised manuscript. Specifically, Tables 4 and 5 have been added to provide more comprehensive information about the flow cytometry results and enhance the clarity of our findings. (Line 324-348)

Table 4. Results of Annexin V-FITC/PI staining for cell apoptosis after B. abortus

Group

Normal cells

The early apoptotic cells (%)

The late apoptotic cells (%)

Control (NI)

84.70±0.39

4.96±0.43

11.94±1.45

S2308-infected

83.40±0.54

1.76±0.43

17.93±1.33

ΔbspF-infected

83.40±0.26

4.36±0.13

14.60±0.46

Table 5. Results of Annexin V-FITC/PI staining for cell apoptosis after B. abortus

Group

Normal cells

The early apoptotic cells (%)

The late apoptotic cells (%)

Control (NI)

73.70±0.37

15.36±1.13

5.7±1.10

S2308-infected

75.40±0.45

4.86±0.23

4.93±0.63

ΔbspF-infected

74.40±0.74

15.66±0.53

5.9±0.23

  1. The Discussion can be extended – as it is now, it seems rather shallow. Also, significant relevant references are missing. Some passages from the Introduction can be transferred to the Discussion.

Reply:We are so sorry for the negligence, and we have added the significant relevant references in the revised manuscript. According to your suggestions, we have added more detailed and deep going introduction and analysis of the mechanism of crotonylation in the discussion section and have transferred some passages from the Introduction to the Discussion in the revised manuscript. (Line 420-434) (Line 475-492) In addition, we have inserted our experiment’s reference significance for studying other intracellular bacteria. (Line 525-527)

Round 2

Reviewer 1 Report

  1. Fig3 A: what does the y-axis label mean? Explain.

Reply:The y-axis is the SSC, lateral corner scattering, and its value represents the cellular granularity. Cell SSC values can be used to compare cell granularity and to group and classify cells.

OK, but this is not shown in revised Fig 3A, label is RA not SSC, therefore I asked. Replace label or inform in Figure legend RA = lateral corner scattering

I mentioned in previous review that in many cases methods lack information. At least provide additional information about Brucella cultivation (section 2.1) and/or references containing the details.

The revision has introduced new mistakes and not all the previous have been corrected. I recommend thorough proofreading before publication. 

Author Response

Comments and Suggestions for Authors

  1. Fig3 A: what does the y-axis label mean? Explain.

Reply:The y-axis is the SSC, lateral corner scattering, and its value represents the cellular granularity. Cell SSC values can be used to compare cell granularity and to group and classify cells.

OK, but this is not shown in revised Fig 3A, label is RA not SSC, therefore I asked. Replace label or inform in Figure legend RA = lateral corner scattering

I mentioned in previous review that in many cases methods lack information. At least provide additional information about Brucella cultivation (section 2.1) and/or references containing the details.

Reply:We apologize for the unclear Y-axis label explanation. Please allow us to elaborate further for better understanding. In Fig. 3A, the value of the Y-axis represents the SSC, or lateral corner scattering, which indicates the cellular granularity. The label on the Y-axis is PI-A, where PI refers to the use of propidium iodide to stain the cells, and A corresponds to the area under the curve of the peak measured by the machine. Thus, the Y-axis is labeled as PI-A. In addition, in the image annotation section of Fig.3A, we also explained the use of propidium iodide (PI) and Annexin V-FITC to stain cells. In order to maintain the accuracy of the graph, we think it is more appropriate to use PI-A as the Y-axis label in Fig. 3A. 

  1. I mentioned in previous review that in many cases methods lack information. At least provide additional information about Brucella cultivation (Section 2.1) and/or references containing the details.

Reply:We are sincerely sorry for our omission. We added Brucella culture methods in Section 2.5 of the manuscript. (Line221-230)

  1. The revision has introduced new mistakes and not all the previous have been corrected. I recommend thorough proofreading before publication. 

Reply:Thanks to the reviewer for pointing out this problem. We have carried out careful proofreading and revision during the revision process.

Reviewer 2 Report

A rushed revision is as annoying as a careless initial version.  Some of the erroneous statements about brucella in the introduction have been erased, but still others stay, indicating that the authors failed to spend some time getting acquainted with the pathogen they are studying: the authors still state that the infection is endemic in 170 countries worldwide. A massive overestimation, a massive error. Please study the epidemiology of brucellosis a bit, write something correct about the main brucella species, it is easy: you can always refer to WHO or OIE. 

On the other hand I am satisfied with the information on crotonylation. 

Author Response

Comments and Suggestions for Authors

  1. A rushed revision is as annoying as a careless initial version.  Some of the erroneous statements about brucella in the introduction have been erased, but still others stay, indicating that the authors failed to spend some time getting acquainted with the pathogen they are studying: the authors still state that the infection is endemic in 170 countries worldwide. A massive overestimation, a massive error. Please study the epidemiology of brucellosis a bit, write something correct about the main brucella species, it is easy: you can always refer to WHO or OIE. 

Reply: We apologize for the mischaracterization of Brucella in the manuscript. In the revised manuscript, we have changed "The spectrum of brucellosis infection is very wide, and brucellosis in clinical recessive or chronic infection is easy to miss and misdiagnose, resulting in widespread prevalence of the disease in more than 170 countries) [2, 3]" to "Brucellosis can occur year-round and is present worldwide, with a particularly severe impact on the health and economy in the Middle East, Africa, and Central and South America". Following your suggestion, we have added an introduction of Brucella spp in the revised manuscript.(Line39-44)

Reviewer 3 Report

The description of the PCRs is still incomplete.
Please provide ALL the details, e.g., the size of the product and the number of bases tested.
The current description does not allow publication, but this can be corrected, after which the manuscript can be accepted.

Author Response

Comments and Suggestions for Authors

1.The description of the PCRs is still incomplete.

Please provide ALL the details, e.g., the size of the product and the number of bases tested.

The current description does not allow publication, but this can be corrected, after which the manuscript can be accepted.

Reply: Thank the reviewer for pointing out this omission. Following your suggestions, we have added primer sequences, target fragment lengths, and restriction enzymes in Table 2. (Line159) In addition, we added the product length of qRT-PCR in Table 4 in the revised manuscript. (Line219)

Table 2. The primer sequences for gene amplification of p53

Primer name

Primer sequences (5`-3`)

Target fragment (bp)

Restriction endonucleases

Flag-p53

Forward:

AAAGAATTCATGGAGGAGCCGC

1182

EcoR I

Reverse:

AAACTCGAGTCAGTCTGAGTCAGGC

Xho I

Flag-p53K351A

Forward:

AGGCCTTGGAACTCGCGGATGCCCAGGCTGG

1182

EcoR I

Reverse:

CCAGCCTGGGCATCCGCGAGTTCCAAGGCCT

Xho I

Table 4. The sequences of primer

Primer names

Primer sequence (5`-3`)

Product length (bp)

p53

Forward:

ATGAGCCGCCTGAGGTTGG

71

Reverse:

CAGTGTGATGATGGTGAGGATGG

Caspase3

Forward:

GTGGAATTGATGCGTGATG

193

Reverse:

TCTCAATGCCACAGTCCAGT

AIF

Forward:

CGGCTCCCAGGCAACTTGTTC

104

Reverse:

GGCACCAGCTCCTACTGTTGATAAG

Bcl-2

Forward:

GGCTACGAGTGGGATGCG

168

Reverse:

TGGACCACAGGTGGCAC

Bax

Forward:

AGAGGATGATTGCCGCC

117

Reverse:

GTGCACAGGGCCTTGAG

Bad

Forward:

CGGAGGATGAGTGACGAGTTTGTG

79

Reverse:

GATCCCACCAGGACTGGAAGACTC

β-actin

Forward:

AAAGACCTGTACGCCAACAC

178

Reverse:

GTCATACTCCTGCTTGCTGAT